# Multi-label Learning with Missing Values using Combined Facial Action Unit Datasets

**Jaspar Pahl** [* 1 2]  **Ines Rieger** [* 1 2]  **Dominik Seuss** [1 2]

## Abstract

Facial action units allow an objective, standardized description of facial micro movements which can be used to describe emotions in human faces. Annotating data for action units is an expensive and time-consuming task, which leads to a scarce data situation. By combining multiple datasets from different studies, the amount of training data for a machine learning algorithm can be increased in order to create robust models for automated, multi-label action unit detection. However, every study annotates different action units, leading to a tremendous amount of missing labels in a combined database. In this work, we examine this challenge and present our approach to create a combined database and an algorithm capable of learning under the presence of missing labels without inferring their values. Our approach shows competitive performance compared to recent competitions in action unit detection.

## 1. Introduction

Action units are distinct facial movements, e.g. "Brow lowerer" or "Cheek raiser". All action units and their characteristics are described in the Facial Action Coding System (FACS) (Ekman et al., 2002). Combining several action units can indicate physical pain (Kunz et al., 2019) or the basic emotions: anger, disgust, fear, happiness, sadness, and surprise (Ekman, 2003). Possible applications of action unit detection include safety-critical areas such as monitoring the driver in driving assisting systems or monitoring pain in the medical field. In these safety-critical areas, a robust detection is essential. One way to achieve this is to train

the model with as many variations of the target object as possible in order to allow for a good generalization.

In this paper we discuss why the combination of datasets for action unit detection is crucial and what kind of missing labels occur in these combined datasets. Furthermore, we describe our methods for training and evaluation with missing labels and show first competitive results.

Supervised deep learning models for image recognition provide state-of-the-art results for action unit detection (Martinez et al., 2017). Deep learning models perform best with a large amount of labeled training data to best cover the target data distribution. But in the field of action units, data is scarce because the process of data collection is expensive. In order to annotate FACS labels, the annotators need to undergo substantial training. The data labeling process takes many times longer than the sequence in question. For inter-rater reliability, each frame is annotated by at least two FACS coders. Furthermore, each dataset has different properties. For each dataset, a different subset of action units is selected to be annotated. In addition, the group of persons in each dataset is usually small, which leads for example to a limited age and ethnic diversity. Therefore, a model trained on a single dataset is severely limited in making robust predictions in real-world scenarios. Thus, it is crucial to combine several datasets to obtain a robust model, as combining datasets with different attributes can reduce bias (Chen et al., 2018). Bias on ethnicity, age or gender has been shown to be a problem in other face-related applications (Grother et al., 2019; Algaraawi & Morris, 2016).

When combining action unit datasets, missing action unit labels occur either due to different subsets of labeled action units or due to facial occlusions. This is different to other domains, where individual missing labels typically occur due to measurement errors or data corruption (Howell, 2019). Consequently, we encounter a considerable higher proportion of missing labels compared to standard missing value scenarios. Our combined database contains 69.07% missing values as seen in Figure 2 in the Appendix A.

Up to now, action unit approaches combining datasets (Chu et al., 2019; Shao et al., 2019; Baltrušaitis et al., 2015) usually only train on action units occurring in all datasets,

---

[*]Equal contribution  [1]Intelligent Systems Group, Fraunhofer Institute for Integrated Circuits IIS, Erlangen, Germany [2]Cognitive Systems Group, University of Bamberg, Bamberg, Germany. Correspondence to: Jaspar Pahl <jaspar.pahl@iis.fraunhofer.de>, Ines Rieger <ines.rieger@iis.fraunhofer.de>.

*Presented at the first Workshop on the Art of Learning with Missing Values (Artemiss) hosted by the $37^{th}$ International Conference on Machine Learning (ICML).* Copyright 2020 by the author(s).

*Table 1.* Properties of the datasets combined in this study. Further explanation can be found in Section 2.1.

| Dataset | Setting | Expression | Format | Subjects | Images | Images/Person |
|---|---|---|---|---|---|---|
| Actor Study | lab | posed | video | 21 | 100,232 | 4,773 |
| Aff-Wild2 | in-the-wild | natural | video | 49 | 196,672 | 4,014 |
| BP4D | lab | natural | video | 41 | 366,954 | 8,950 |
| CK+ | lab | posed | video | 123 | 10,733 | 87 |
| EmotioNet manual | in-the-wild | natural | images | 24,597 | 24,597 | 1 |
| UNBC | lab | natural | video | 25 | 48,397 | 1,936 |

thus circumventing the problem of the missing labels. Other approaches use probabilistic methods to approximate the missing labels by exploiting domain knowledge (Niu et al., 2019; Li et al., 2019; Peng & Wang, 2018; Wu et al., 2014). Due to the high amount of missing labels, we only train on ground-truth labels and not on probabilistically inferred labels. We do not fully drop samples with any missing label. Instead, our approach is to mask the single missing labels and occlude them in training and evaluation, using only ground-truth annotated labels.

In the next Section 2, we present an overview of the different properties of a range of datasets to show the diversity of the annotations in order to give an impression on the proportion of missing labels. Furthermore, we give a short description on how we build our meta-database. In Section 3, we present our methods for training and evaluation of the models trained with missing labels and show the results. We conclude with Section 4.

## 2. Data

### 2.1. Datasets

As explained in introductory Section 1, combining datasets with different recording settings and different annotation content is beneficial for action unit detection. In this paper, we have included the following datasets: Actor Study (Seuss et al., 2019), Aff-Wild2 (Kollias & Zafeiriou, 2018a), BP4D (Zhang et al., 2014), CK+ (Lucey et al., 2010), the manually annotated subset of EmotioNet (Fabian Benitez-Quiroz et al., 2016), and UNBC (Lucey et al., 2011). Upon comparison, it is clear that these datasets have been recorded in vastly different settings and therefore sample from different distributions (Table 1). The different properties are explained in the following.

Data recorded in laboratory settings generally has a uniform background, standardized viewing angles on the face, consistent lighting conditions, and pre-selected participants for the study. The latter implies potentially strong bias towards age, ethnicity, and sometimes sex of the participants, leading to bias in the visual appearance of persons in the dataset.

In-the-wild datasets such as the EmotioNet or Aff-Wild2 in contrast have varying background and lighting. Due to often being sampled from web-searches containing data from all around the world, in-the-wild datasets show considerably lower bias to visual appearance of the displayed persons.

Another key difference is in whether the action units displayed are acted or occur naturally. Acted action unit displays tend to differ from their natural versions due to the actors intentionally displaying the action units separately and performing comparably strong expressions. Furthermore, displaying specific action units on demand is a training intensive task, which many, particularly non-professional, actors are not trained for; leading to further deviation.

Some datasets offer only single images of an action unit display, others show full video sequences. Among the latter, some annotate only one value for an action unit label for the whole sequence, others provide a frame level coding. Video based datasets offer a higher count in images than image-based ones, but many of those images are very similar.

While action unit coding is a standardized process by definition in the Action Unit Manual (Ekman et al., 2002), practice shows there are significant differences in how datasets are annotated for action units. Due to action unit coding being a time-intensive task requiring professional staff, most datasets only mark action units they require for their specific task. Table 3 in Appendix A shows binary action unit appearance for each dataset. Furthermore, each group of annotators leaves out different information for the action unit in question. Very few datasets have full intensity coding, some define their own, reduced bins for intensity, most do not annotate it at all. Standardized action unit coding should contain onset, peak, and offset of each action unit, but in reality this information is to varying degrees left out.

### 2.2. Meta-Database

Due to vastly different coding schemes described above and different data formats provided for annotations, we have defined our own meta-database. For this procedure we manually define which action units are annotated for each dataset based on the associated paper, providing the unknown action

units. Original annotation files are first checked manually for any obvious errors or missing data, and then read in and transferred to a unified labeling system. Those labels are finally saved in a python pandas (Pandas Dev Team, 2020; Wes McKinney, 2010) data frame which includes links to the original image files, meta information, and labels. This procedure allows us to read labels from a single, standardized source and to fine-tune our training dataset as described in Section 3.1.

A visualization of the action unit distribution concerning the annotated and missing labels in the merged dataset can be seen in Figure 2 in Appendix A.

# 3. Experiment

## 3.1. Methods

We use customized loss and metric functions, since we need to omit missing labels in the loss function and in the metric: When an unknown label occurs in the ground truth label vector, we delete this vector row in the predicted label vector and the ground truth label vector. This process is computed for each class separately.

For evaluating the model's performance, we use the F1 macro score and the accuracy. Accuracy by itself could be misleading for an imbalanced multi-label classification problem as it does not take the true to false ratio for one class into account. The F1 macro score on the other hand compensates for an imbalanced ratio of displayed and non-displayed action unit labels and furthermore the imbalance between classes. Therefore, we select the best model from one training by the average of the F1 macro score (Eq. 1b) and accuracy (Eq. 1c). The variable $N$ in Eq. 1b and Eq. 1c stands for the number of action units. The notation $|\{0, 1\}$ in Eq. 1a and Eq. 1c means that only not displayed and displayed action units are included, but not the unknown.

$$F1 = \frac{2 \cdot precision_{|\{0,1\}} \cdot recall_{|\{0,1\}}}{precision_{|\{0,1\}} + recall_{|\{0,1\}}} \quad (1a)$$

$$F1_{macro} = \frac{1}{N} \left( \sum_i^N F1_i \right) \quad (1b)$$

$$Acc = \frac{1}{N} \left( \sum_i^N \frac{Correct\_predictions_{|\{0,1\},i}}{All\_predictions_{|\{0,1\},i}} \right) \quad (1c)$$

We use the F1 macro score as a loss function (Eq. 2) by making it differentiable using probabilities instead of discrete values. The F1 loss offers advantageous properties when dealing with imbalanced multi-label classification. As in the metrics before, we calculate the loss only based on annotated values, omitting the unknown labels. In batch training however, the occurrence of labels for each action

unit can consequently vary because of the missing labels. This can distort the calculation of the loss function in comparison to a training with no missing labels. We accept this disadvantage for now, because batch training smoothes the convergence of a model.

$$L = 1 - F1_{macro} \quad (2)$$

## 3.2. Training

We use the neural network architecture VGG16 (Simonyan & Zisserman, 2015) pretrained with the ImageNet dataset (Deng et al., 2009) from the open source deep library Keras (Chollet et al., 2015). For finetuning we apply a learning rate of $0.0001$ and use the Adam optimizer with AMSGrad (Reddi et al., 2019). The activation function of the output layer is a sigmoid function. We train our model for 30 epochs with a batch size of 256. We first freeze all convolutional layers and train two fully connected layers with each 256 units and a subsequent ReLu activation function and dropout layer with a rate of $0.5$. Then we pick the best model of this training and finetune the last convolutional block.

We train on our combined database including all datasets described in Section 2.1. In order to ensure stable training we concentrate on action units of at least $20,000$ occurrences. Occurrence refers to an action unit being displayed on a face. Since action units are heavily imbalanced in their nature, we balance the displayed action units of the selected subset with a multi-label balance optimizer (Rieger et al., 2020). This leads to the distribution shown in Figure 1. We use color information of each image and normalize pixel values in a range of [0,1]. We leave out facial images with only unknown annotations for the selected action units. This results in 570,053 training images and 104,545 for testing. Table 2 shows the amount of displayed action units for training and testing and our model's results on the testing dataset.

## 3.3. Results

The results in Table 2 show that our trained model performs competitively well. We cannot compare our approach directly to other approaches, but there are two recent challenges for action unit detection for orientation: (1) Affective Behavior Analysis in-the-wild challenge (Kollias et al., 2020) at FG 2020, and (2) EmotioNet Challenge (ENC, 2020) at CVPR 2020. The ABAW challenge evaluates on eight action units on the Aff-Wild2 dataset, the top contestant reaches an F1 macro score of 0.31 (Deng et al., 2020). The Aff-Wild2 dataset is also part of our combined database. We include all of Aff-Wild2's action units except for AU20, due to it being insufficiently represented according to our threshold. The EmotioNet challenge evaluates on 23 action units on a unpublished part of the EmotioNet manual dataset,

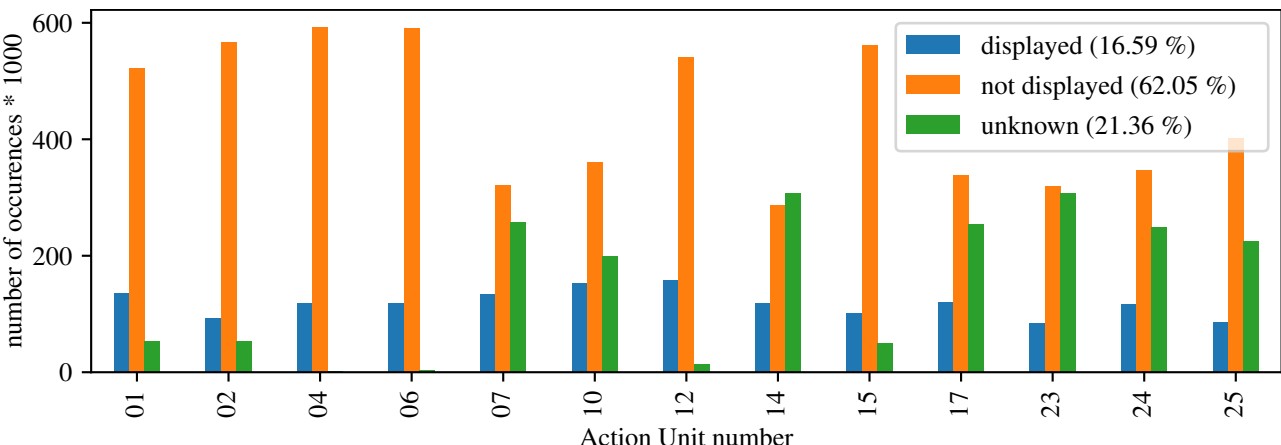

*Figure 1.* Distribution of all action units in the merged database after removal of images with only unknown values for action units and balancing of the resulting set according to Section 3.2. Number of occurrences are shown for the categories displayed (a positive annotation for an action unit), not displayed (a explicit negative annotation for an action unit), and unknown (no information on status of an action unit).

*Table 2.* This table shows the amount of displayed action units for training (Train) and testing (Test). The model is evaluated on the testing dataset. The dataset used for weighting the F1 macro score is identified by ✓.

| AUs | Train | Test | F1 |
|---|---|---|---|
| AU01 | 107,007 | 20,944 | 0.69 |
| AU02 | 78,461 | 5,528 | 0.42 |
| AU04 | 94,986 | 14,150 | 0.59 |
| AU06 | 95,539 | 21,212 | 0.63 |
| AU07 | 107,283 | 20,124 | 0.73 |
| AU10 | 121,313 | 22,056 | 0.80 |
| AU12 | 129,915 | 22,550 | 0.77 |
| AU14 | 94,422 | 15,010 | 0.62 |
| AU15 | 83,580 | 6,099 | 0.33 |
| AU17 | 98,892 | 11,388 | 0.53 |
| AU23 | 65,769 | 5,970 | 0.35 |
| AU24 | 94,006 | 7,781 | 0.54 |
| AU25 | 71,048 | 5,727 | 0.59 |
| F1 macro | - | - | 0.58 |
| Weighted F1 macro | - | ✓ | 0.65 |
| Weighted F1 macro | ✓ | - | 0.61 |

whereby the top three participants reach a F1 macro score of 0.44 to 0.55. The manually annotated, fully published part of the EmotioNet dataset is also part of our combined database. We include all of its action units, except AU07, AU14 and AU23 with the same threshold reasoning. These results of two major challenges can therefore serve as indication that our F1 macro score of 0.58 is competitively high.

This leads to the assumption that the model learns to generalize action unit features from the different data distributions of the combined database. However, there is still a difference in performance between the action units. By calculating the weighted F1 macro score using the action unit occurrences of the training and testing dataset, it is apparent that the number of occurrences has an influence on the result in comparison to the unweighted F1 macro score. Looking at the weighted F1 scores, the influence in the distribution of the testing dataset is higher than the influence in the training dataset. We assume the reason to be that the variance of distribution in the testing dataset is higher in contrast to the training dataset.

## 4. Conclusion

Deep learning models have revolutionized image recognition in the course of the last decade. A key element of their success is the availability of sufficiently large annotated databases for training. In this paper we show how to enable deep learning based action unit detectors to use combined databases with high numbers of missing labels. This allows for the transfer of the success factor of large databases to the domain of action units, which has notorious problems with insufficient data.

While it is difficult to directly compare the results to other approaches trained on differing data distributions or less data, our approach is very competitive when compared to the most recent action unit challenges. It even supersedes them in F1 macro score performance on our combined database, which contains highly difficult in-the-wild datasets.

## Acknowledgements

We thank Robert Obermeier for support in data management and Sarah Stenz for proof reading. Parts of this work have been funded by the Federal Ministry of Education and Research, grant no. 01IS18056A (TraMeExCo), other parts by German Research Foundation, grant no. GA 2485/3-1 (PainFaceAnalyzer).

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

# A. Supplementary Materials

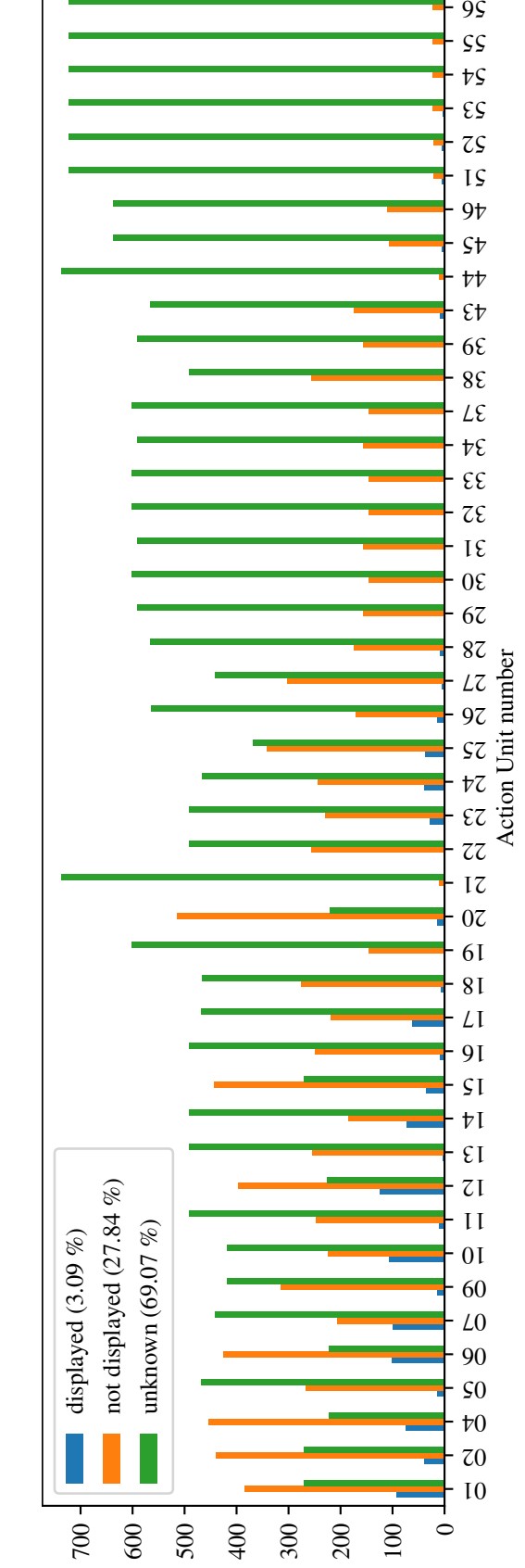

*Table 3.* Action unit annotations of the datasets combined in this study, described in Section 2.1

*Figure 2.* The distribution of all action units in the merged database. The number of occurrences are shown for the categories displayed (a positive annotation for an action unit), not displayed (an explicit negative annotation for an action unit), and unknown (no information on status of an action unit). Due to combining datasets, there is a significant amount of missing labels.