# OpenReview forum: "Multi-label Learning with Missing Values using Combined Facial Action Unit Datasets"
_ICML.cc/2020/Workshop/Artemiss — ICML Artemiss 2020_

### Official Review · AnonReviewer1 · 2020-06-30
**Interesting problem that could lead to very nice discussions**

**Confidence:** 3
**Rating:** 7

**Review:**

The authors explore the missing data problem in the context of Facial action analysis.

Facial action analysis is basically a multi-label classification problem where features are images of faces, and labels are facial micro movements.

The goal of the authors is to combine a lot of combined facial action unit datasets in order to train a single deep model on them. One big issue is that different data sets may look at different facial micro movements, which can be seen as a missing data problem.

The authors describe the problem quite clearly, and show that simply ignoring the missing values while training can lead to a very competitive model.

I think that the problem is very interesting, and certainly deserves to be discussed at this workshop. My main issue is that I don't fully understand what's the loss function used for training. Indeed, it looks like the authors suggest to use a loss based on the F1 score. But the F1 score is not a differentiable loss function, similarly to the 0-1 loss but unlike, e.g. cross-entropy. This should definitely be clarified in the final version.

Discussing under which missingness assumptions the approach suggested makes sense (e.g. missing completely at random) would be interesting.

---

### Decision · Program_Chairs · 2020-07-02

**Decision:**

Accept

**Comment:**

We are very happy to inform you that your paper has been accepted for the Artemiss workshop. We will contact you soon to inform you about the details concerning the format of your presentation at the workshop, and the camera-ready version deadline. Please take into account the referee's comments to write the camera-ready version.